# Urinary incontinence among pregnant women in Southern Brazil: A population-based cross-sectional survey

**Hsu Yuan Ting**[1]*, **Juraci A. Cesar**[2]

1 Postgraduate Program in Health Sciences, Federal University of Rio Grande (FURG), Rio Grande, Rio Grande do Sul, Brazil, 2 Postgraduate Program in Public Health, Postgraduate Program in Health Sciences, Federal University of Rio Grande (FURG), Rio Grande, Rio Grande do Sul, Brazil

* hsuriogrande@gmail.com

**Data Availability Statement:** The data underlying the results presented in the study are available from "Perinatal 2016 Rio Grande - urinary incontinence" https://dx.doi.org/10.17504/protocols.io.bfxzjpp6.

## Abstract

Urinary incontinence (UI) is a common condition that causes significant harm to the well-being and quality of life of pregnant women. This cross-sectional population-based study aimed to estimate the prevalence and identify factors associated with the occurrence of UI during pregnancy in women living in the municipality of Rio Grande (RS), Southern Brazil, between January 1 and December 31 of 2016, and included all puerperae living in this municipality that had a child in one of the two local maternity hospitals. The previously trained interviewers used a single standardized questionnaire, within 48 hours after delivery to retrieve information on maternal demographic, behavioral and reproductive/obstetric history, as well as socioeconomic status of the household and care received during pregnancy and childbirth. The multivariate analysis followed a previously defined hierarchical model using Poisson regression with robust variance adjustment and prevalence ratio (PR) as a measure of effect. As a result, 2,716 puerperae were identified, of which 2,694 (99.2%) participated in this study. The prevalence of urinary incontinence in the gestational period was 14.7% (95%CI: 13.4%-16.1%). After adjusted analysis, the likelihood of UI occurring varied significantly as per women's characteristics. For example, the PR for the occurrence of UI among women over 30 years of age was 2.05 (95% CI: 1.39–3.01) compared to adolescents. In two other groups of women who had their first pregnancy before the age of 20 or after the age of 30, the PR for UI was 1.36 (95% CI: 1.04–1.76) and 1.59 (95% CI: 1.01–2.51), respectively, when compared to those who became pregnant for the first time between 20 and 29 years of age. Finally, in two other groups of women, namely, those who reached 90 kg and over at the end of pregnancy and those who performed regular physical exercise and reported frequent urinary urgency, the PR was 2.49 (95% CI: 1.74–3.57), and 2.90 (95% CI: 2.10–4.00) compared to those who did not exercise and did not report urinary urgency, respectively. The authors concluded that UI showed a high prevalence in the study population. The identified risk factors can be well administered at primary health care level. The recommendation of regular physical exercise in pregnancy must be reviewed and better investigated with more robust designs because of possible facilitators for the occurrence of UI in this period.

**Funding:** The author(s) received no specific funding for this work.

**Competing interests:** The authors have declared that no competing interests exist.

## Introduction

Urinary incontinence (UI) is defined as any involuntary loss of urine, and the prevalence increases with advancing age [1]. Its occurrence ranges from 40% to 60% among women, and from 10% to 20% among men [2].

This higher occurrence among women is generally due to their reproductive life [3]. Hormonal changes, enlargement of the uterus, pelvic floor changes during gestation, and trauma suffered during delivery lead to involuntary loss of urine [4–6].

UI during gestation is a significant predictor for its presence in subsequent pregnancies and at a later age [7], which makes it a chronic disease with a substantial deterioration of the quality of life, whether due to discomfort, the need for regular personal hygiene, or insecurity, among others. At a later age, UI leads to isolation, which favors depression and more severe psychiatric conditions [8,9].

While very prevalent, UI has been rarely studied at the population level in Brazil. The few available studies are performed with a minimal number of pregnant women, usually from a single health service, without any type of representativeness at the population level [10–12]. Besides preventing the establishment of actions and programs due to lack of knowledge of the real magnitude of the problem, this situation hinders prevention at the primary level of health care, which contributes to the persistence and severity of this disease, increases suffering, and deteriorates the quality of life of these women, especially in the gestational period and in older age.

This study aims to measure the prevalence and to identify factors associated with the occurrence of UI in the gestational period among puerperae living in the municipality of Rio Grande (RS), Southern Brazil, during 2016.

## Materials and methods

Data shown here derive from the 2016 Perinatal Study, which is part of a series of triennial cross-sectional surveys, held in the municipality of Rio Grande since 2007. These evaluations aim to monitor the quality of gestation and delivery care provided in this municipality. The research protocol was submitted and approved by the Health Research Ethics Committee (CEPAS) of the Santa Casa de Misericórdia of Rio Grande (file Nº 30/2015). Data confidentiality, voluntary participation, and the possibility of leaving the study at any time without the need for justification were assured.

Pregnant women should reside in the municipality of Rio Grande (in urban or rural areas), must have had a child in one of the two local maternity hospitals (Santa Casa de Misericórdia or the University Hospital) from January 1 to December 31, 2016, with a birth weight equal to or greater than 500 grams, or with at least 20 weeks of gestational age, to be included in this survey.

The cross-sectional design was used, and mothers were approached only once in the maternity ward within 48 hours after delivery.

Concerning the sample size, two calculations were performed, namely, one to estimate the prevalence, and the other to identify associated factors; in both cases, we added 10% of possible losses, which means, women who did not want to participate in the study or who left the hospital before being invited to participate. In the first sample, the study should include at least 2,334 puerperae, and regarding the second, 2,680 mothers. We used significance level of 95% [13].

The outcome of this study was established by the event of urinary incontinence in the gestational period evaluated by a positive response to the following question: "During this gestation, did you ever lose urine unintentionally?" The information about this study was collected

through a single, pre-coded questionnaire applied by interviewers previously trained using tablets and the REDCap (Research Electronic Data Capture) application [14].

Three previously trained interviewers collected data daily. Puerperae were asked to participate in the study, and then the Informed Consent Form (ICF) was signed. Questionnaires were uploaded daily through the REDCap Web platform, and data consistency was checked and immediately corrected. The consistency analysis included the categorization of variables and frequency verification, and was performed using Stata statistical package version 12.0 [15].

Approximately 10% of the interviews were retaken in order to evaluate the quality of the data collected, which was done later by telephone or home visit, where a summary questionnaire was applied. The Kappa concordance index ranged from 0.68 to 0.89.

Results were expressed by the prevalence, and as a measure of effect, we employed the prevalence ratio (PR), whose interpretation is similar to that provided to relative risk, in cohort studies, or odds ratio, in case-control studies. We also used a 95% confidence interval (95% CI), and the p-value of the trend test and the Wald test for heterogeneity [16]. Crude and adjusted analysis was performed using Poisson regression, with robust adjustment for variance [17]. The adjusted analysis was conducted from a previously defined four-level hierarchical model [18]. This adjusted analysis aims to eliminate the effect of confounding factors, that is, it separates the unique and exclusive effect of the variable in question on the endpoint, eliminating the effect of other variables that are not being tested.

These levels were used to determine the order of entry of the variables in the model. At the first level, demographic and socioeconomic variables (age, skin color, living with a partner, schooling, household income and paid work during pregnancy) were included; the reproductive variable (age at the first pregnancy) was entered at the second level; variables related to prenatal and delivery care (number of prenatal consultations, trimester of onset of consultations, delivery type) and nutritional status (weight at the end of pregnancy) were added at the third level. The fourth and last level included variables related to habits and behavior (smoking, coffee consumption, and regular physical activity in the gestational period) and morbidity (urinary urgency). The outcome was the event of urinary incontinence during pregnancy.

All the variables were taken to the multivariate model, and those with a value of $p \leq 0.20$ were maintained. Analyses were conducted in the Stata 12.0 program, and the level of significance was 95%.

## Results

The National Live Births Information System [19] and the Mortality Information System evidenced 2,716 births whose mothers lived in the municipality of Rio Grande. Of this total, 2,694 were interviewed, revealing a respondent rate of 99.2% (or a loss of 0.8%).

Table 1 shows the distribution of all puerperae by the main characteristics studied. About 14.7% (95% CI: 13.4–16.1) of women reported having urinary incontinence. Of these, 52.3% had stress incontinence, 18.4% urge incontinence, and 29.3% mixed. Also, 8.8% of them started urinary loss in the first trimester of gestation, 27% in the second and 64.2% in the third trimester, and all of them had UI until the end of gestation.

Table 2 shows the crude and adjusted analysis of the prevalence of the studied variables, and we found five factors associated with the UI event. In the adjusted analysis, the PR for puerperae aged 30 years or older was 2.05 (95% CI: 1.39–3.01) compared to adolescents; mothers who had their first pregnancy aged 30 years or older, or before the age of 20, had PR = 1.59 (95% CI: 1.01–2.51) and 1,36 (95% CI: 1.04–1.76), respectively, compared to those who had their first pregnancy at 20–29 years. In this same table, we found that the higher the weight at the end of pregnancy, the higher the PR for UI occurrence. PR for the occurrence of UI among

**Table 1. Prevalence of urinary incontinence according to some characteristics of puerperal residents in Rio Grande, Brazil, 2016.**

| Variables | Total (n) |
|---|---|
| Maternal age (years) | |
| 12–19 | 16.9% (456) |
| 20–29 | 49.7% (1,340) |
| ≥30 and over | 33.3% (898) |
| Maternal skin color | |
| White | 67.0% (1,806) |
| Brown | 22.7% (610) |
| Black | 10.3% (278) |
| Living with partner | 83.6% (2,252) |
| Household income in minimum wages | |
| 0–0.9 | 8.5% (215) |
| 1–3.9 | 69.8% (1,775) |
| ≥4 | 21.7% (553) |
| Maternal schooling (full years) | |
| 0–8 | 36.7% (990) |
| 9–11 | 39.8% (1,071) |
| ≥12 | 23.5% (633) |
| Engaged in paid work during pregnancy | 45.9% (1,237) |
| Age (years) at first pregnancy | |
| 12–19 | 60.2% (924) |
| 20–29 | 35.2% (540) |
| ≥30 | 4.6% (71) |
| Number of visits | |
| 0–5 | 15.7% (422) |
| 6–11 | 72.5% (1,954) |
| ≥12 | 11.8% (318) |
| Started prenatal care in the first trimester | 78.9% (2,094) |
| Delivery type | |
| Vaginal | 45.8% (1,234) |
| Cesarean | 54.2% (1,460) |
| Weight (kg) at the end of pregnancy | |
| 40–69.9 | 28.7% (754) |
| 70–79.9 | 26.2% (688) |
| 80–89.9 | 22.1% (581) |
| ≥90 | 23.0% (607) |
| Drank coffee during pregnancy | 33.9% (912) |
| Smoked during pregnancy | 20.6% (341) |
| Engaged in regular exercise during pregnancy | 5.7% (154) |
| Had urinary urgency | |
| Never | 61.8% (1,665) |
| Sometimes | 32.7% (881) |
| Often | 5.5% (148) |
| Prevalence of urinary incontinence | 14.7% (396) |
| Total | 100% (2,694) |

those weighing 90 kg or more was 1.63 (95% CI: 1.17–2.27) compared to those who had a weight lower than 70 kg at the end of gestation. Finally, regular physical exercise during

**Table 2. Prevalence of urinary incontinence by category and crude and adjusted analyses as per the hierarchical model.** Rio Grande (RS), Brazil, 2016.

| Level | Variables | Prevalence of urinary incontinence | Prevalence ratio (CI 95%) | |
|---|---|---|---|---|
| | | | Crude | Adjusted |
| I | Maternal age (years) | | p<0.00 | p<0.001 |
| | 12–19 | 8.3% | 11.0 | *1.0 |
| | 20–29 | 14.3% | 01.72 (1.23–2.40 | 01.64 (1.13–2.38 |
| | ≥30 | 18.5% | )2.22 (1.59–3.10) | )2.05 (1.39–3.01) |
| | Skin color | | p = 0.09 | p = 0.26 |
| | White | 15.5% | 51.0 | 51.0 |
| | Brown/Black | 13.1% | 00.84 (0.69–1.03) | 00.89 (0.72–1.10) |
| | Household income in minimum wages | | p = 0.02 | p = 0.349 |
| | 0–0.9 | 9.3% | 81.0 | *1.0 |
| | 1–3.9 | 14.9% | 01.60 (1.04–2.47 | 01.38 (0.89–2.15 |
| | ≥4 | 17.2% | )1.85 (1.17–2.91) | )1.35 (0.83–2.20) |
| | Maternal schooling (full years) | | p = 0.058 | p = 0.986* |
| | 0–8 | 12.9% | 1.00 | 1.00 |
| | 9–11 | 14.9% | 1.15 (0.93–1.43) | 1.02 (0.81–1.28) |
| | ≥12 | 17.2% | 1.33 (1.05–1.69) | 1.02 (0.77–1.36) |
| | Living with partner | | p = 0.061 | p = 0.600 |
| | Yes | 15.3% | 1.00 | 1.00 |
| | No | 11.8% | 0.77 (0.59–1.01) | 0.92 (0.68–1.25) |
| | Engaged in paid work during pregnancy | | p = 0.021 | p = 0.383 |
| | Yes | 16.4% | 1.00 | 1.00 |
| | No | 13.3% | 0.81 (0.67–0.97) | 0.91 (0.74–1.12) |
| ii | Age (years) at first pregnancy | | p = 0.035 | p = 0.031* |
| | 12–19 | 16.8% | 1.19 (0.93–1.54) | 1.36 (1.04–1.76) |
| | 20–29 | 14.1% | 1.00 | 1.00 |
| | ≥30 | 25.4% | 1.80 (1.15–2.83) | 1.59 (1.01–2.51) |
| iii | Number of visits | | p = 0.033 | p = 0.098* |
| | 0–5 | 12.1% | 1.00 | 1.00 |
| | 6–11 | 14.6% | 1.21 (0.91–1.60) | 1.12 (0.74–1.68) |
| | ≥12 | 18.9% | 1.56 (1.10–2.20) | 1.53 (0.94–2.49) |
| | Trimester of onset of prenatal care visits | | p = 0.061 | p = 0.605 |
| | First | 15.4% | 1.00 | 1.00 |
| | Second and third | 12.2% | 0.79 (0.62–1.01) | 0.92 (0.66–1.28) |
| | Delivery type | | p = 0.035 | p = 0.981 |
| | Vaginal | 13.1% | 1.00 | 1.00 |
| | Cesarean | 16.0% | 1.22 (1.01–1.47) | 1.00 (0.79–1.27) |
| iii | Weight (kg) at the end of pregnancy | | P<0.001 | p = 0.016* |
| | 40–69.9 | 11.1% | 1.00 | 1.00 |
| | 70–79.9 | 14.1% | 1.27 (0.96–1.66) | 1.09 (0.76–1.57) |
| | 80–89.9 | 16.2% | 1.45 (1.10–1.91) | 1.27 (0.89–1.82) |
| | ≥90 | 19.1% | 1.72 (1.32–2.22) | 1.63 (1.17–2.27) |

(*Continued*)

**Table 2.** (Continued)

| Level | Variables | Prevalence of urinary incontinence | Prevalence ratio (CI 95%) | |
|-------|-----------|-----------------------------------|---------------------------|---|
| | | | Crude | Adjusted |
| iv | Drank coffee during pregnancy | | P = 0.030 | p = 0.632 |
| | Never drank | 12.6% | 1.00 | 1.00 |
| | Drank | 15.8% | 1.25 (1.02–1.53) | 1.06 (0.83–1.36) |
| | Smoked during pregnancy | | p = 0.102 | p = 0.219 |
| | No | 14.3% | 1.00 | 1.00 |
| | Yes | 17.6% | 1.23 (0.96–1.58) | 1.21 (0.89–1.66) |
| | Engaged in regular exercise during pregnancy | | p<0.001 | p<0.001 |
| | No | 13.7% | 1.00 | 1.00 |
| | Yes | 31.2% | 2.27 (1.76–2.93) | 2.49 (1.74–3.57) |
| | Had urinary urgency | | p<0.00 | p<0.001* |
| | Never | 10.6% | 11.0 | 1.00 |
| | Sometimes | 18.4% | 01.74 (1.43–2.12) | 1.74 (1.36–2.22) |
| | Often | 39.2% | )3.71 (2.90–4.73) | 2.90 (2.10–4.00) |

* Wald trend test.

pregnancy and reporting frequent urinary urgency showed a PR of 2.49 (95% CI: 1.74–3.57) and 2.90 (95% CI: 2.10–4.00) compared to those who did not exercise and did not report urinary urgency, respectively.

## Discussion

This study found a prevalence of UI in the gestational period of 14.7%. It also showed that the likelihood of this disease, even after adjustment, is significantly higher among pregnant women who got pregnant or had a child in adolescence, who weighed 90 kg or more at the end of gestation, who performed regular physical exercises and who reported frequent urinary urgency during the gestational period.

The prevalence of UI found in this survey is low compared to other studies, ranging from 15% [20] to 71% [11]. This enormous discrepancy arises from different characteristics of the participants, such as the inclusion of nulliparous alone, and the diagnostic criteria used, often based on a single question of the event of involuntary urine loss [21–23].

Maternal age is an inexorable marker of the occurrence of UI. The more advanced the age, the higher its prevalence. This may be due to the loss of innervation and the gradual reduction in the contraction capacity of muscle fibers and increased permeability of the urethral sphincter [24], which leads to a lower pressure of its closure [25], resulting in the involuntary loss of urine. A recent systematic review conducted in the European population found OR = 1.4 (95% CI: 1.3–1.5) for UI among those 35 years of age or older compared to younger age [20]. A similar result was found in this study. Mothers aged 20–29 years and 30 years or older showed a PR for UI of 1.64 (95% CI: 1.13–2.38) and 2.05 (95% CI: 1, 39–3.01), respectively. This evidences the strength of the variable age as a risk factor for this condition.

Maternal age at the time of the first gestation was also significantly associated with the probability of UI in the studied population. A similar finding was found in a cross-sectional study conducted in Norway with about 11,000 women [26]. In this study, women with gestation before 25 years of age showed a prevalence of UI of 23% versus 28% among those who had a child at a later age (p> 0.001). In this study from Rio Grande, having a child at 20–29 years showed the lowest risk of UI compared to those who had a child before the age of 20 or after

the age of 30, which is probably due to pelvic floor trauma at younger ages and loss of muscle fibers and urethral sphincter pressure at later ages.

In this study, weighing over 90 kg at the end of gestation showed PR = 1.63 (95% CI: 1.17–2.27) compared to the others. The relationship between body mass index (BMI) and UI is usually directly proportional. This risk factor is already well established [27,28], which may not only be due to the relationship between weight and height but also that the gestational period shows an increased bone density and peripheral edema, and is also influenced by hormonal factors and fetal weight. This set of factors may be responsible for increased weight gain being a significant risk factor for UI.

In Rio Grande, even after adjustment, regular physical exercise appeared as a predisposing factor to UI concerning the other pregnant women, which is even more worrying because obstetricians usually recommend physical exercise, called "fitness", to pregnant women during pregnancy. They claim the benefits of this practice to maternal-fetal health and, because of this, are included in several guidelines as a healthy measure for gestation [29].

It is well known that, even at a young age, elite female athletes have a higher prevalence of UI. This prevalence can affect about half of them [30]. A Norwegian study conducted among academy instructors, including Pilates and Yoga teachers, found a prevalence of UI of 26.4% among instructors with a mean age of 32.8 years (± 8.3). This rate is very similar to that observed in the general female population [31]. These data suggest that physical exercises can overload the pelvic floor, thus increasing the likelihood of UI. In the case of pregnant women, who already have an overload, this is even more serious. Hence the need for this indication to be very well-defined, mentioning exactly which exercises, their frequency, and at what time of gestation they can be performed. Otherwise, this indication may favor UI. It should be noted, however, that when the exercise is directed to the pelvic floor musculature training (PFMT), it has been effective in reducing the occurrence of UI, with RR = 0.71 (95% CI: 0, 54–0.95) compared to those who did not perform this type of training [32]. Also, on this subject, a meta-analysis showed that PFMT, by reducing labor time, especially in the first and second stages reduces pelvic floor trauma and, therefore, can prevent the occurrence of UI [33].

About 40% of pregnant women in this study reported urinary urgency, that is, a sudden sensation that makes it very hard to postpone urination. Of these, about six percent referred to this condition as "very frequently". The PR for the probability of UI of this group among those who did not report urinary urgency was very high at 2.90 (95% CI: 2.10–4.00). The association between urgency and incontinence is so close that it was set in the next upper level of the end-point, showing the relevance of this condition to the occurrence of UI.

Urinary urgency is a widespread problem among pregnant women. A Brazilian study found a prevalence of this condition in 44% of the participants [34]. The main reason for urinary urgency is the increased blood volume and the effect of circulating hormones during pregnancy [35], but this should be clarified better.

We should consider that this is a cross-sectional study when interpreting these results. Therefore, caution should be exercised when evaluating some associations because the exposure and outcome variables were collected at the same time. However, we could not find in the literature a more robust design that has worked with such a significant number of pregnant women like this one.

## Conclusions

This population-based study showed that urinary incontinence is a common disease among pregnant women, and also confirmed the findings of other investigations for maternal age and urinary urgency as a risk factor for this ailment. It suggested that the variables "age at first

gestation", "weight over 90 kg at the end of pregnancy", and "engaging in regular physical exercise in this period" may be associated with this disease as well. As a result, we recommend that professionals providing prenatal care pay attention to these factors, and suggest that these three variables be included in future research on this topic when adopting a more robust design such as a cohort.

## Author Contributions

**Conceptualization:** Hsu Yuan Ting, Juraci A. Cesar.

**Data curation:** Hsu Yuan Ting.

**Formal analysis:** Juraci A. Cesar.

**Funding acquisition:** Hsu Yuan Ting.

**Investigation:** Hsu Yuan Ting.

**Methodology:** Juraci A. Cesar.

**Project administration:** Juraci A. Cesar.

**Writing – original draft:** Hsu Yuan Ting, Juraci A. Cesar.

**Writing – review & editing:** Hsu Yuan Ting, Juraci A. Cesar.

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
