## [Decision Letter · Decision Letter 0]

11 Mar 2020

PONE-D-19-28652

Urinary incontinence among pregnant women in Southern Brazil a population-based cross-sectional survey

PLOS ONE

Dear Mr. TING,

Thank you for submitting your manuscript to PLOS ONE. After careful consideration, we feel that it has merit but does not fully meet PLOS ONE’s publication criteria as it currently stands. Therefore, we invite you to submit a revised version of the manuscript that addresses ALL the points raised during the review process.

SPECIFIC ACADEMIC EDITOR COMMENTS: Your manuscript was handled by two expert reviewers. Although interest was found in your study, there were some comments that arose that need addressing. These comments include, but are not limited to several statements throughout the manuscript that need clarification or abbreviating along with methodology and statistics that need to be better explained.

We would appreciate receiving your revised manuscript by Apr 25 2020 11:59PM. To enhance the reproducibility of your results, we recommend that if applicable you deposit your laboratory protocols in protocols.io, where a protocol can be assigned its own identifier (DOI) such that it can be cited independently in the future. For instructions see: http://journals.plos.org/plosone/s/submission-guidelines#loc-laboratory-protocols

We look forward to receiving your revised manuscript.

Kind regards,

Frank T. Spradley

Academic Editor

PLOS ONE

Journal Requirements :

2) We note that you have stated that you will provide repository information for your data at acceptance. Should your manuscript be accepted for publication, we will hold it until you provide the relevant accession numbers or DOIs necessary to access your data. If you wish to make changes to your Data Availability statement, please describe these changes in your cover letter and we will update your Data Availability statement to reflect the information you provide.

3) Your ethics statement must appear in the Methods section of your manuscript. If your ethics statement is written in any section besides the Methods, please move it to the Methods section and delete it from any other section. Please also ensure that your ethics statement is included in your manuscript, as the ethics section of your online submission will not be published alongside your manuscript.

Reviewers' comments:

Reviewer's Responses to Questions

**Comments to the Author**

1. Is the manuscript technically sound, and do the data support the conclusions?

Reviewer #1: Yes

Reviewer #2: Yes

2. Has the statistical analysis been performed appropriately and rigorously? 

Reviewer #1: Yes

Reviewer #2: Yes

3. Have the authors made all data underlying the findings in their manuscript fully available?

Reviewer #1: Yes

Reviewer #2: Yes

4. Is the manuscript presented in an intelligible fashion and written in standard English?

Reviewer #1: Yes

Reviewer #2: No

5. Review Comments to the Author

Reviewer #1: As stated on p. 15, urinary incontinence (UI) is common among pregnant women. The proposed article claims that a positive association exist between pregnant women and UI that is measurable within a define set of inclusion criteria as defined on p. 10. The study is significant given the prevalence and occurrence as stated in the article. The study target population involves a population-based cross-sectional survey of mothers who have given birth within a defined gestational period and/or birth weight (20 weeks gestational age and/or ≥500g).

As mentioned in the article, “regarding sample size, two calculations were performed to estimate prevalence and factors associated with UI in pregnant mothers for the proposed study.” Data obtained from retrospective 2016 perinatal surveys was used to identify association factors and estimate prevalence ratio. Adjustments for eventual losses and adjustments for potential confounders were taken into account and well defined.

Informed consent forms were appropriately signed and provided to birth mothers prior to implementing the study. In addition, a signed copy of the informed consent was retained at the hospital.

In terms of data quality and integrity, data management involved follow up telephone calls and home visits to account for consistency of data. As mentioned in the article, approximately 10% of the interviews were retaken to "evaluate the quality of the data collected."

The study power involved the use of multivariate analysis that followed a hierarchical model to link prevalence of the UI to associated variables with adjusted expected outcomes. Based on this model, the study determined the prevalence of UI in gestational period, lowest observed rate of UI in the study population, the highest frequency of urinary urgency among other variables as outlined in the study.

P. 12 The study outcome was clearly defined.

P.19 Reference list clearly demonstrates literature review for relevant systematic review, meta-analysis, REDcap, and essential tools needed for medical statistics on public health related issues.

Reviewer #2: Dear Author!

This is a cross-sectional study aiming at estimating the prevalence of and risk factors for urinary incontinence during pregnancy among women living in Rio Grande, Brasil.

My comments for you are listed below:

General:

Why did you not separate the conditions stress urinary incontinence and urgency urinary incontinence, which are quite different in etiology?

Abstract:

- UI is a frequent pathology = UI is a frequent condition

- Measure the prevalence = estimate the prevalence

Background:

- “…and increases with age” =”…and the prevalence increases with advancing age”

- This is a journal read mainly by health care professionals, it is therefore unnecessary to explain that UI may be experienced by both sexes. What is essential, however, is that you use the background section to focus on why it is important to examine UI in your population, when similar studies have been performed in other populations. Are you f.i. suspecting that women in Rio Grande would be especially prone to UI due to (I am just guessing) heavy rural work or young age at first delivery?

- Rewrite the following, since in its present form it seems like all factors should be present in the same patient: “The probability of occurrence of UI, even after adjustment, was significantly higher among those who were older than 30 years old at current pregnancy, whose first pregnancy was before the age of 20 or after 30, who reached the end of gestation weighing 90 kg or more, who exercised regularly during pregnancy and who reported frequent urinary urgency during the gestational period.”

Methods:

- First paragraph-shorten or remove

- Why cut-off at 20 weeks?

- Power-calculation hard to follow, and must be better described: especially the following part “…ratio between exposed and unexposed/exposed of 15/85, prevalence of disease among the unexposed of 9.8% and 1.6 risk ratio..”

- eventual losses = eventual lost to follow-up/ lost data or deaths?

- Remove repeated statements like, like p.4 “…the following question: "During this gestation, did you ever lose urine unintentionally?" A positive response would mean puerperae had UI during this gestational period.”

- Condense the text from “Three interviewers collected data….” to “…

was performed using Stata statistical package version 12.0. “

- When were the 10% of the interviews retaken? Did you consider recall bias? What happened when the data were inconsistent, which questionnaire was used- the first or the second?

- Explain the advantage of using Poisson regression models in non-longitudinal data instead of regular linear or logistic regression?

- Remove inadequate racial expressions like skin color, white, brown, black etc. (also in results section and tables)

Results:

- Incredible that more than 99% participated voluntarily in a study, how was this possible?

- Do not repeat extensively in text what is in presented the tables

Discussion:

- Modify strong expressions such as “this is due to the loss of….”, rather use “this may be due to..” or similar.

- Great age= old age/ advanced age

Conclusions:

- Avoid the expression causally associated, since to detect causal effects from a cross-sectional study would be very hard.

6. PLOS authors have the option to publish the peer review history of their article (what does this mean?). If published, this will include your full peer review and any attached files.

Reviewer #1: Yes: Barbara A. Wilson

Reviewer #2: No

---

## [Author Response · Author response to Decision Letter 0]

9 Apr 2020

We attached the Response to Reviewers document.

---

## [Decision Letter · Decision Letter 1]

28 Apr 2020

PONE-D-19-28652R1

Urinary incontinence among pregnant women in Southern Brazil a population-based cross-sectional survey

PLOS ONE

Dear Mr. TING,

Thank you for submitting your manuscript to PLOS ONE. After careful consideration, we feel that it has merit but does not fully meet PLOS ONE’s publication criteria as it currently stands. Therefore, we invite you to submit a revised version of the manuscript that addresses the points raised during the review process. ALL of the reviewers' comments and concerns must be addressed in your revision.

We would appreciate receiving your revised manuscript by Jun 12 2020 11:59PM. To enhance the reproducibility of your results, we recommend that if applicable you deposit your laboratory protocols in protocols.io, where a protocol can be assigned its own identifier (DOI) such that it can be cited independently in the future. For instructions see: http://journals.plos.org/plosone/s/submission-guidelines#loc-laboratory-protocols

We look forward to receiving your revised manuscript.

Kind regards,

Frank T. Spradley

Academic Editor

PLOS ONE

Reviewers' comments:

Reviewer's Responses to Questions

**Comments to the Author**

1. If the authors have adequately addressed your comments raised in a previous round of review and you feel that this manuscript is now acceptable for publication, you may indicate that here to bypass the “Comments to the Author” section, enter your conflict of interest statement in the “Confidential to Editor” section, and submit your "Accept" recommendation.

Reviewer #1: All comments have been addressed

Reviewer #2: All comments have been addressed

2. Is the manuscript technically sound, and do the data support the conclusions?

Reviewer #1: Yes

Reviewer #2: Yes

3. Has the statistical analysis been performed appropriately and rigorously? 

Reviewer #1: Yes

Reviewer #2: Yes

4. Have the authors made all data underlying the findings in their manuscript fully available?

Reviewer #1: Yes

Reviewer #2: Yes

5. Is the manuscript presented in an intelligible fashion and written in standard English?

Reviewer #1: Yes

Reviewer #2: Yes

6. Review Comments to the Author

Reviewer #1: Abstract revisions were appropriately applied.

As stated on p. 15, urinary incontinence (UI) is common among pregnant women.  The proposed article claimed that a positive association exists between pregnant women and UI that is measurable within a defined set of inclusion criteria as defined on p. 10.  The study is significant given the prevalence and occurrence as stated in the article.  The study target population involves a population-based cross-sectional survey of mothers who have given birth within a defined gestational period and/or birth weight (20 weeks gestational age and/or ≥500g).

As mentioned in the article, “regarding sample size, two calculations were performed to estimate prevalence and factors associated with UI in pregnant mothers for the proposed study.”    Data obtained from retrospective 2016 perinatal surveys was used to identify association factors and estimate prevalence ratio.  

Informed consent forms were appropriately signed and provided to birth mothers prior to implementing the study.   In addition, a signed copy of the informed consent was retained at the hospital. 

In terms of data quality and integrity, data management involved follow up telephone calls and home visits to account for consistency of data.  As mentioned in the article, approximately 10% of the interviews were retaken to "evaluate the quality of the data collected."  

The study power involved the use of multivariate analysis that followed a hierarchical model to link prevalence of the UI to associated variables with adjusted expected outcomes.   Based on this model, the study determined the prevalence of UI in gestational period, lowest observed rate of UI in the study population, the highest frequency of urinary urgency among other variables as outlined in the study.  

P. 12 The study outcome was clearly defined.

P.19 The reference list clearly demonstrates literature reviews on public health related issues.

Reviewer #2: Dear Author!

Thank you for the response to my comments and your effort to improve the paper.

I still have a few comments regarding the methods section:

-You do not have to over-explain the power calculation in a statistical sense, however I am asking you to explain or reference your assumptions. Where did the ratio of unexposed/exposed to UI come from? I suppose the power calculation was performed before you started the study and not performed post-hoc?

- The expression “losses” should be briefly explained: for instance: ..losses, including women who did not want to participate in the study or who left the hospital before being invited to participate….

- I still find the entity “skin color” problematic. Since the Brazilian population is ethnically mixed, the genetic importance of skin color is dubious. Because skin color does not have any relevance for your results, and in order to avoid offending readers, I would recommend the removal of this entity. However, I leave the decision to the editor.

7. PLOS authors have the option to publish the peer review history of their article (what does this mean?). If published, this will include your full peer review and any attached files.

Reviewer #1: No

Reviewer #2: No

---

## [Author Response · Author response to Decision Letter 1]

4 May 2020

Dear Reviewer #2

Here are your comments and our answers, thank you for considering our efforts for improving this paper:

- You do not have to over-explain the power calculation in a statistical sense, however I am asking you to explain or reference your assumptions. Where did the ratio of unexposed/exposed to UI come from? I suppose the power calculation was performed before you started the study and not performed post-hoc?

Answer: We made new adjustments to not over-explain the power calculation. However, for answering the question, the calculation of the sample size, we used data published in other studies and a pilot study carried out before the beginning of data collection. The formula used to calculate the sample parameters was this: A/(B+C*E), where: A=Prevalence of outcome; B=Proportion of non-exposed; C=Proportion of exposed; D=Prevalence of outcome among non-exposed; E=Prevalence ratio (Dean AG, Dean JA, Coulombier D, Brendel KA, Smith DC, Burton AH, Dicker RC, Sulliven K, Fagan RF, Arner TG. Epi-Info, Version 6.02: A Word Processing, Database, and Statistics Program for Epidemiology on Microcomputers. Atlanta (GA): Centers of Disease Control and Prevention; 1994). (Line 93 -98)

- The expression “losses” should be briefly explained: for instance: ..losses, including women who did not want to participate in the study or who left the hospital before being invited to participate….

Answer: Done. Line 95 -96.

- I still find the entity “skin color” problematic. Since the Brazilian population is ethnically mixed, the genetic importance of skin color is dubious. Because skin color does not have any relevance for your results, and in order to avoid offending readers, I would recommend the removal of this entity. However, I leave the decision to the editor.

Answer: We reiterate that we do not evaluate ethnicity, but only the color of the skin, for the reasons already explained in a previous message. In Brazil, saying that someone has white, brown or black skin color is not offensive and has been widely used in scientific articles. I kindly ask you to look for articles by Victora CG, Barros FC and other Brazilian authors, where you can see the wide use of this name. If this does not convince the reviewer, then we leave it to the editor to decide.

---

## [Decision Letter · Decision Letter 2]

26 May 2020

Urinary incontinence among pregnant women in Southern Brazil: a population-based cross-sectional survey

PONE-D-19-28652R2

Dear Dr. TING,

We are pleased to inform you that your manuscript has been judged scientifically suitable for publication and will be formally accepted for publication once it complies with all outstanding technical requirements.

With kind regards,

Frank T. Spradley

Academic Editor

PLOS ONE

Additional Editor Comments (optional):

Reviewers' comments:

Reviewer's Responses to Questions

**Comments to the Author**

1. If the authors have adequately addressed your comments raised in a previous round of review and you feel that this manuscript is now acceptable for publication, you may indicate that here to bypass the “Comments to the Author” section, enter your conflict of interest statement in the “Confidential to Editor” section, and submit your "Accept" recommendation.

Reviewer #2: All comments have been addressed

2. Is the manuscript technically sound, and do the data support the conclusions?

Reviewer #2: Yes

3. Has the statistical analysis been performed appropriately and rigorously? 

Reviewer #2: Yes

4. Have the authors made all data underlying the findings in their manuscript fully available?

Reviewer #2: Yes

5. Is the manuscript presented in an intelligible fashion and written in standard English?

Reviewer #2: Yes

6. Review Comments to the Author

Reviewer #2: Dear author!

Regardless of what may be expressed in other Brazilian papers, I find the use of skin colour as an entity hard to swallow.However, I leave the decision upon this issue to the Editor. I have no further comments.

7. PLOS authors have the option to publish the peer review history of their article (what does this mean?). If published, this will include your full peer review and any attached files.

Reviewer #2: No

---

## [Editor Report · Acceptance letter]

29 May 2020

PONE-D-19-28652R2 

Urinary incontinence among pregnant women in Southern Brazil: a population-based cross-sectional survey 

Dear Dr. TING:

I am pleased to inform you that your manuscript has been deemed suitable for publication in PLOS ONE. Congratulations! Your manuscript is now with our production department. 

With kind regards,

on behalf of

Dr. Frank T. Spradley 

Academic Editor

PLOS ONE